# Molecular Identification of Pro-Excitogenic Receptor and Channel Phenotypes of the Deafferented Lumbar Motoneurons in the Early Phase after SCT in Rats

**DOI:** 10.3390/ijms231911133

**Published:** 2022-09-22

**Authors:** Benjun Ji, Bartosz Wojtaś, Małgorzata Skup

**Affiliations:** 1Group of Restorative Neurobiology, Nencki Institute of Experimental Biology, 02-093 Warsaw, Poland; 2Laboratory of Sequencing, Nencki Institute of Experimental Biology, 02-093 Warsaw, Poland

**Keywords:** hyperexcitability, spasticity, spinal cord injury, AMPAR, NMDAR, GlyR, GABA_A_R, KCC2, 5-HTRs

## Abstract

Spasticity impacts the quality of life of patients suffering spinal cord injury and impedes the recovery of locomotion. At the cellular level, spasticity is considered to be primarily caused by the hyperexcitability of spinal α-motoneurons (MNs) within the spinal stretch reflex circuit. Here, we hypothesized that after a complete spinal cord transection in rats, fast adaptive molecular responses of lumbar MNs develop in return for the loss of inputs. We assumed that early loss of glutamatergic afferents changes the expression of glutamatergic AMPA and NMDA receptor subunits, which may be the forerunners of the developing spasticity of hindlimb muscles. To better understand its molecular underpinnings, concomitant expression of GABA and Glycinergic receptors and serotoninergic and noradrenergic receptors, which regulate the persistent inward currents crucial for sustained discharges in MNs, were examined together with voltage-gated ion channels and cation-chloride cotransporters. Using quantitative real-time PCR, we showed in the tracer-identified MNs innervating extensor and flexor muscles of the ankle joint multiple increases in transcripts coding for AMPAR and 5-HTR subunits, along with a profound decrease in GABA_A_R, GlyR subunits, and KCC2. Our study demonstrated that both MNs groups similarly adapt to a more excitable state, which may increase the occurrence of extensor and flexor muscle spasms.

## 1. Introduction

Spinal cord injury (SCI) causes severe neuronal impairment, resulting in a loss of muscle function and sensation below the injury site [1]. Consequently, the most profound symptom of patients with SCI is the loss of motor ability leading to paraplegia or tetraplegia, depending on the level of injury. Another common complication of SCI is spasticity which significantly impacts the quality of life of patients suffering SCI and impedes locomotion recovery. 

In humans, early clinical signs of spasticity (clonus, hypertonia, spasms) in patients with SCI during hospitalization were observed as early as 30 days following injury, associated with decreased long-term functional outcomes [2,3]. Spasticity caused by SCI is often marked by a slow increase in excitation and over-activity of both flexors and extensors, with reactions possibly occurring many segments away from the stimulus [4]. In experimental SCI in rodents predominantly used for SCI modeling, the onset of spasticity can be seen as early as one-week postinjury [5,6,7]. After a mild contusive SCI at the lumbar L2 level in adult mice, signs of spasticity of hindlimb muscles were observed 1-week postinjury [6]. Similarly, in a tail-spasticity model in an adult rat with a complete spinal cord transection (SCT) at the sacral S2 segment, the first signs of spasticity of the tail were observed 1-week postinjury, which was sustained at 2–3 weeks and gradually worsened until 2 months [5]. In rats with SCT at the thoracic T11 level, the development of hindlimb spasticity examined at 3, 8, and 16 weeks postlesion, showed the highest level at 3 weeks [8]. Hsieh and co-workers, who quantified the changes of tonic stretch reflexes (muscle tone) following a spinal cord hemisection at the thoracic T8 level in rats, showed that the muscle tone dropped immediately after spinal shock and then gradually increased to reach a peak exceeding the control values around 3 weeks postinjury [9]. Muscle tone remained at least 75% of the peak value up to the end of an 8-week observation period. Taken together, these data indicate that the spasticity, with hypertonus in flexor and extensor muscles, clonus, and hyperreflexia, begins to develop around the first week postinjury and reaches the peak after 3 weeks, varying slightly between studies using different rodent SCI models. Although the models differ from the human SCI in anatomy, neurofunction, and pathophysiological response, they provide valuable insights for preclinical therapies for patients with SCI [10,11]. 

At the cellular level, spasticity is considered to be primarily caused by the hyperexcitability of spinal α-motor neurons (MNs) within the spinal stretch reflex (H-reflex) circuit [12,13,14] because these MNs directly innervate muscles and drive body movements. Due to the loss of descending supraspinal pathways and segmental changes in local circuits after SCI, the excitability of MNs, which is an intrinsic ability of neurons to generate action potentials (AP) determined by the function and abundance of neurotransmitter receptors and ion channels in neuronal membranes [15], is altered [16,17]. Specific changes in gene expression, translation, and biophysical properties of receptors, ion channels [18,19,20,21,22], and potassium-chloride transporter [23] in the MNs after SCI have been postulated to contribute to a decreased threshold and increased probability of the generation of APs, leading to hyperexcitability of MNs and subsequent spasticity [12].

The importance of dendritic spine dysgenesis in the development of spasticity is also emerging [14]. In a recent study, Benson and co-workers [6] provided the first evidence for a mechanistic relationship between MN dendritic spine dysgenesis, hyperexcitability, and SCI-induced spasticity. Till now, evidence for dendritic dysgenesis and molecular changes in neurotransmission-related molecules in MNs in spinal animals has been obtained mostly for the late postlesion phase (3–12 weeks), when hyperexcitability is well established. However, susceptibility of MNs to the altering synaptic activity in response to a loss of input signaling or physical training in spinal rats, manifested by stabilizing or remodeling of dendritic architecture, has been shown as early as five days after SCT [24]. These observations point to the potential of MNs to adaptive but also maladaptive plasticity in the acute postlesion phase. Adaptation might be under the control of glutamatergic receptors if postlesion regulation of MN dendritic architecture resembles a sensitive period in early postnatal life [25,26].

In the present study, we hypothesized that after the acute phase, when MN excitability plummets, fast molecular responses of lumbar MNs develop in response to the loss of inputs. We assumed that early loss of glutamatergic sensory afferents, shown by us recently [27], changes the expression of ionotropic excitatory glutamatergic (AMPA and NMDA) receptors, which may be the forerunners of the developing spasticity of the hindlimb muscles. Our recent data, which showed a decrease in transcript levels of *Grin1* NMDA and *Gria2* AMPA receptor subunits after spinalization and nerve implantation, support this assumption [27]. Here we aimed to disclose the direction and extent of changes in their transcript levels in the hindlimb MNs in the subacute phase after SCI (second-week postinjury) when the spinal shock is resolved and the spasticity starts to develop [5,6,9]. To better understand the molecular underpinnings of developing hyperexcitability early postlesion, we sought to investigate also whether SCI leads to a concomitant decrease of transcript expression of the inhibitory GABA and Glycinergic receptors and altered expression of serotoninergic (5-HT) and noradrenergic (NA) receptors in response to the loss of descending sources of 5-HT and NA, respectively [28,29]. We primarily examined the 5-HT_2A_, 5-HT_2B_, 5-HT_2C,_ and α(_1A_) adrenergic receptor subtypes, which regulate the persistent inward currents (Ca and Na PICs) crucial for sustained discharges in response to synaptic stimuli in MNs [18,30,31,32,33], and whose constitutive activity developing late after SCT in a tail spasticity model, contributes to the emergence of spasms of the tail muscles [21,22]. We also characterized gene expression of selected voltage-gated ion channels, and two types of neuronal cation-chloride cotransporters reported to regulate GABA-mediated MN signaling [23,34,35,36].

We used complete spinal cord transection at the thoracic T11/12 level, applied in our previous studies [27,37,38], to address the lumbar motor nuclei and hindlimb muscles. The evidence that the H-reflex circuit is impaired after SCT to a different extent in MNs innervating extensor and flexor muscles operating at the ankle joint [27,37,39] prompted us to design the experiment in which we characterized separately molecular responses in pools of the lumbar MNs innervating ankle extensor (Gastrocnemius lateralis; GL) and flexor (Tibialis anterior; TA) muscles. For that purpose, we analyzed transcript levels using quantitative real-time PCR (qPCR) in the tracer-identified, isolated MNs belonging to these two functional groups.

Our results provide new insight into the mechanisms of MN excitability developing in the subacute phase below the site of SCT, involving a multiple increase in transcript coding for AMPAR and 5-HTR subunits and a profound decrease in GABA_A_R, GlyR subunits, and KCC2. Our study demonstrates that both MNs groups similarly adapt to a more excitable state, which may increase the occurrence of extensor and flexor spasms.

## 2. Results

### 2.1. MNs at Lumbar 3–6 Segments Show Normal Morphology after SCT at Low Thoracic Level

The experimental procedures applied to identify and characterize GL and TA MNs are shown in Figure 1A,B (see more details in Materials and Methods). Both types of MNs were visualized under a laser microdissection (LMD) microscope in fluorescence mode (Figure 1C). GL MNs were located within the gray matter of the lumbar 4-6L4-6 segment, whereas most TA MNs were located in the gray matter of the L3-4 segment, consistent with other studies [34,37,40,41].

One profound feature of SCI is neuronal loss at and around the injury site. At the injury epicenter of severe SCI, neurons degenerate, undergoing necrosis or apoptosis [43,44,45,46]. Significant loss of MNs has been observed by 48 h postinjury at 4.0 mm from the epicenter [43]. Since in the present study the spinal MNs located at 13–15 mm from the T11 injury epicenter were the subject of gene expression analysis, it raised the necessity to identify whether there is a neuron loss or shrinkage which may affect the collection of the material and transcript levels. Therefore, during microdissection, the numbers of traced GL and TA MN sections were counted on each horizontal L3–6 section (Figure 1D). Our results showed that the number of MNs of a given type was comparable between intact and spinalized rats (GL: SCT = 94% CN, *p* = 0.53; TA: SCT = 98% CN, *p* = 1.00; Mann-Whitney *U* Test). In the spinal group, the tracer deposits filled the cell bodies and the dendritic processes of MNs, indicating that tracer transport to MNs and its maintenance was not impaired compared to the control group. In addition, the morphology of the MN perikarya in the lumbar segment was similar between CN and SCT rats. No apoptotic profiles or shrunken neurons were found in Hoechst and Nissl-stained sections from SCT rats (Figure 1C and Appendix A). Thus, we conclude that there were no obvious signs of degeneration of MNs located at the lumbar L3–6 segment. 

### 2.2. Steady-State mRNA Abundance of Receptor and Ion Channel Subunits in GL and TA MNs of the Intact Rats

In the present study, forty genes (Appendix A) were analyzed. The selected genes encode the major subunits of the excitatory (AMPA, NMDA), inhibitory (GABA_A_, Gly), modulatory (mGlu, 5-HT, and NA) receptors, ion channels (Na^+^, Ca^2+,^ and K^+^), and K^+^ -Cl^−^ cotransporters (KCCs). Additionally, the expression of the gene coding for the ADAR2 enzyme that catalyzes the Q/R editing of the GluA2 AMPAR subunit was analyzed.

In intact rats, all groups of genes showed similar expression patterns in the GL and TA MNs (Figure 2 and Appendix A). Except for *Gria3* (Wilcoxon test, *p* = 0.04), *Gabrb3* (Wilcoxon test, *p* = 0.02), and *Htr2c* (Wilcoxon test, *p* = 0.04), the transcript level of other genes did not significantly differ between GL and TA MNs. Among them, genes encoding for the majority of subunits of inhibitory receptors: *Gabrb3* (GABA_A_ β3), *Gabrg2* (GABA_A_ γ2), *Glra1* (Gly α1), and *Glrb* (Gly β), as well as *Kcnma1* (KCa1.1) channel demonstrated the highest transcript level, followed by *Scn8a* (Nav1.6), *Scn1a* (Nav1.1), *Grin2a* (GluN2A), *Grin2b* (GluN2B), *Kcnn2* (SK2), and *Gabra2* (GABA_A_ α2) and *Gabra3* (GABA_A_ α3) receptor subunits. The remaining sets of genes displayed relatively low transcript levels in both pools of examined MNs. Among receptors, the subunits which combine to form subclasses of inhibitory GABA_A_R and GlyR were more highly expressed than subunits of other neurotransmitter receptors, both in GL and TA MNs.

### 2.3. SCT Differentially Alters Transcript Level of Subunits of Glutamatergic AMPA, NMDA, and mGlu Receptors

AMPARs are mainly tetramers consisting of GluA1, GluA2, GluA3, and GluA4 subunits, encoded by *Gria1*, *Gria2*, *Gria3,* and *Gria4* genes, respectively [46,47]. The majority of AMPA receptors in the adult brain contain GluA2 subunits; GluA2 can be enzymatically edited at the Q/R site, changing a glutamine (Glu) to an arginine (Arg) within the ion pore, which renders GluA2 virtually Ca^2+^ impermeable [48]. Consequently, GluA2-containing AMPAR is impermeable to Ca^2+^, while GluA2-lacking AMPAR is Ca^2+^ permeable. In the brain, in the basal physiological state, the vast majority of AMPARs consist of GluA2/GluA3 and GluA1/GluA2 subunit combinations, with a small population of GluA1/GluA1 homomers [47]. Based on the quantitative comparisons of our data shown in Figure 3, we suggest that a similar relationship takes place in the lumbar spinal cord motoneurons: *Gria2* (GluA2) and *Gria3* (GluA3) transcript levels were much higher than *Gria1* (GluA1) and *Gria4* (GluA4) transcript levels (Figure 3), suggesting that GluA2/GluA3 is the major subunit combination of AMPAR in GL and TA MNs. The expression level of *Gria4,* much lower than of the other subunits of AMPAR, is consistent with the observation that *Gria4* is primarily expressed in early postnatal development, while *Gria1–3* are expressed in most neurons in the brain in adulthood [47].

SCT led to a significant increase in the transcript level of the *Gria1* (GluA1) subunit (*p* = 0.01 both in GL and TA MNs), a decrease in the *Gria3* (GluA3) subunit (*p* = 0.004 in GL MNs, *p* = 0.002 in TA MNs), while *Gria2* (GluA2) subunit was unchanged in either of the two MN groups (Figure 3). The result suggests that the proportion of GluA1/GluA2 subunit combinations in AMPARs may be increased, while that of GluA2/GluA3 subunit combinations is decreased. Further speculation of the consequences of that direction of change is that more GluA1/GluA1 homomers may be formed after injury. Because AMPARs containing the GluA1/GluA1 subunit combination are Ca^2+^ permeable, the switch from GluA2-containing AMPARs to GluA2-lacking AMPARs increases the possibility of MN enrichment in synaptic Ca^2+^-permeable AMPARs (CP-AMPAR) resulting in membrane depolarization and sustained AP firing in MNs. The appearance and dynamic regulation of CP-AMPARs in MNs postinjury might enable these modified receptors to contribute to signaling via calcium influx [48].

In the brain, the most common NMDAR, a tetrameric complex also contains two GluN1 and two GluN2 (typically one GluN2A and one GluN2B) subunits [49]. We show that SCT led to decreased transcript levels of *Grin1* (GluN1)*, Grin2b* (GluN2B), *Grin2c* (GluN2C), and *Grin2d* (GluN2D) subunits, while it did not alter *Grin2a* (GluN2A) transcript level neither in GL nor TA MNs (Figure 3). Consequently, the general decrease of NMDAR subunit gene expression after SCT may lead to decreased NMDAR signaling in GL and TA MNs. 

Metabotropic glutamatergic receptors mGluRs play a role in the modulation of excitability of neurons after ligand binding. In MNs of intact rats, we demonstrate that *Grm1a* (mGluR1A) showed higher mRNA expression than *Grm5* (mGluR5) (Figure 3), which is in line with a result of an early in situ hybridization study by Tölle’s group [50]. We found that SCT resulted in an increased transcript level of group I mGluRs. Namely, the transcript level of *Grm1a* (mGluR1A) was increased profoundly both in GL and TA MNs after SCT, with *Grm5* (mGluR5) elevated in GL MNs but not in TA MNs (Figure 3).

### 2.4. SCT Downregulates Transcript Level of GABAergic and Glycinergic Receptor Subunits and KCC Transporters Similar to in GL and TA MNs

Inhibition of MNs is primarily mediated by GABAergic and glycinergic neurotransmission, which is regulated by intraneuronal chloride (Cl^−^) homeostasis. We report that SCT leads to profound decreases in the transcript levels of all examined subunits of inhibitory receptors GABA_A_R and GlyR both on GL and TA MNs (Figure 4). 

Next, we examined the expression level of genes encoding K^+^ -Cl^−^ co-transporters: *Slc12a4* (KCC1), *Slc12a5* (KCC2), and *Slc12a6* (KCC3), which primarily extrude chloride ions out of the neurons regulating the intraneuronal Cl^−^ homeostasis. Among them, *Slc12a5* (KCC2) displayed the highest transcript level in intact animals (Figure 4), confirming that this neuron-specific KCC2 is the major Cl^−^ extruder, which was found of comparable abundance in GL and TA MNs.

SCT caused a significant downregulation of *Slc12a5* (KCC2) and *Slc12a6* (KCC3) genes, while *Slc12a4* (KCC1) gene expression was upregulated (Figure 4). The predominant decrease of the transcript level of KCC2 and KCC3, if followed by changes in respective proteins, may indicate decreased abundance and function, resulting in the accumulation of high concentrations of Cl^−^ in the cytoplasm of GL and TA MNs. If so, a depolarizing effect instead of an inhibitory response may be expected after ligand binding to GABA_A_ and Gly receptors. 

### 2.5. SCT Downregulates Transcript Level of Nav1.6, KCa1.1(BK), and SK2 Channels Similar to in GL and TA MNs

Action potentials (APs) are directly initiated by the opening of voltage-gated (VG) Na^+^ and Ca^2+^ channels and are negatively regulated by the activity of Ca^2+^-activated K^+^ (SK and BK) channels. SK channels mediate the outward K^+^ currents and generate the medium afterhyperpolarization (mAHP), which reduces the firing frequency of APs, leading to decreased excitability of neurons [51]. Thus, increased persistent Na^+^ and Ca^2+^ inward current (PIC) and decreased K^+^ outward current increase the probability of APs generation and, subsequently, the excitability of neurons. Therefore, we examined whether SCT affects the transcript levels of major Na^+^, Ca^2+,^ and K^+^ channels related to the generation of APs.

Among the four examined VG Na^+^ channels, *Scn1a* (Nav1.1) and *Scn8a* (Nav1.6) displayed the highest and comparable transcript levels in GL and TA MNs, followed by the moderate *Scn9a* (Nav1.7) expression (Figure 5). *Scn3a* (Nav1.3) was expressed at an extremely low level in both MN groups, in line with a study showing differences in expression signal in the lumbar spinal cord using in situ hybridization [52]. After SCT, *Scn8a* (Nav1.6) was significantly downregulated in GL and TA MNs. *Scn9a* (Nav1.7) was selectively decreased in GL MNs, while *Scn1a* (Nav1.1) and *Scn3a* (Nav1.3) were kept unaltered.

The expression of two VG Ca^2+^ channels, *Cacna1b* (Cav2.2) and *Cacna1d* (Cav1.3), in the intact rats, was two orders of magnitude lower than that of VG Na^+^ channels and was unaffected by SCT. Conversely, the expression of small-conductance Ca^2+^-activated K^+^ (SK) channels was markedly changed (Figure 5). In GL and TA MNs of intact animals, *Kccn2* (SK2) displayed the highest transcript level, followed by the *Kccn3* (SK3), while the transcript level of *Kccn1* (SK1) was extremely low, suggesting that SK2 undergoes extensive turnover and plays the major roles in these MNs. These results align with the study showing that SK2 protein is expressed in all α-MNs of the lumbar segment in the rat, whereas SK3 is expressed preferentially in small-diameter α-MNs [53]. Regulation of gene expression of SK2 and SK3 in response to the lesion was in the opposite direction: a significant decrease of the transcript level of *Kccn2* (SK2), parallel in both groups of MNs, was accompanied by increased *Kcnn3* (SK3) level (Figure 5). Highly abundant *Kcnma1* transcripts of large-conductance (KCa1.1, BK) channel, which can integrate changes in intracellular calcium and membrane potential contributing to the regulation of outward K^+^ currents, were significantly downregulated both in GL and TA MNs (Figure 5).

### 2.6. SCT Alters Transcript Levels of 5-HT Receptors but Not NA Receptors

Finally, we examined the expression of serotoninergic 5-HT receptors and noradrenergic NA receptors (Figure 6), which have been demonstrated to play crucial roles in modulating the firing of MNs in the spinal cord [30,54]. Examination of transcript levels of the 5-HTR subunits after the lesion revealed that the expression of *Htr1a* (5-HTR1A) and *Htr2b* (5-HTR2B) subunits, which are abundant in normal rats, markedly increased, and a degree of their up-regulation was similar in both MNs groups. These changes were accompanied by an increase of the *Htr2a* (5-HTR2A) subunit in GL MNs only, and a selective decrease of the *Htr2c* (5-HTR2C) subunit in TA MNs (Figure 6).

Among the examined NA receptor subunits, *Adra1a* (NAα1A) and *Adra1d* (NAα1D) expression were comparable, whereas *Adra1b* (NAα1B) displayed very low expression, suggesting that *Adra1a* and *Adra1d* predominate in GL and TA MNs. The expression of none of these subunits was changed (Figure 6).

### 2.7. Correlation Analyses of Gene Expression in GL and TA MNs after SCT

Correlation analyses were carried out using classical Pearson correlation tests for every pairwise combination of all genes (corr.test R package). The heat-mapped correlograms were produced using the corrplot package under R, with a scale from −1 to 1. The correlograms were generated for GL and TA MNs from CN and SCT animals to search for the differences in the co-expression pattern of all gene pairs.

First, the correlation analysis indicated that the correlated patterns of gene expression differ between GL MNs and TA MNs from CN animals (Figure 7). Second, the patterns of correlated gene expression are changed after SCT both in GL and TA MNs (Figure 7). For instance, in GL MNs of CN rats, there was a clear negative or positive correlation between the expression of the NMDAR GluN2A subunit and ion channels subtypes (Figure 7, box A), but these correlations were lost after SCT (Figure 7, box A’). Similarly, there were relationships among Na^+^ and Ca^2+^ channel subtypes in GL MNs of CN rats (Figure 7, box B), which were substantially weakened or disappeared after SCT (Figure 7, box B’). The same observation concerns TA MNs (Figure 7, box C and C’).

The correlograms with hierarchical clustering were also analyzed (Figure 8). With the function of hierarchical clustering, the genes with similar expression patterns group together as a cluster, indicating these genes are co-regulated in either CN or SCT groups of rats. The results clearly showed there are different clusters containing different genes in GL and TA MNs from CN animals. Furthermore, SCT substantially altered the distribution and density of these clusters (Figure 8). For instance, in GL MNs of CN rats, genes *Gria2*, *Grin1*, *Glrb,* and *Gabra2* are positively correlated in one cluster (Figure 8, box A). However, SCT dispersed the four genes into four different clusters (Figure 8, box A’). Similar effects of SCT also occur in TA MNs.

These results suggest the set of genes coding for neurotransmission-related membrane proteins undergoes differential regulation in different functional types of MNs under the influence of spinal cord injury already in the second postlesion week. 

## 3. Discussion

Our previous studies showed that SCT leads to a long-term decrease (5–6 weeks) in markers of synaptic nerve endings abutting motoneurons [38]. A decrease in synaptophysin labeling (by 18%) and in zinc staining (by 26%) in terminals surrounding large neurons of the ventral horn in lumbar L3/4 segments was in line with subsequent works, which revealed a decrease in the number of cholinergic C-boutons [37], and glutamatergic terminals around identified groups of MNs [27,55]. These symptoms of MN denervation prompted us to ask whether MNs adapt to altered signaling, leading to molecular changes underlying their altered excitability. In the present study, by focusing on the transcriptional alterations within the selected groups of MNs, we demonstrated that complete SCT at a low thoracic level leads to early, significant changes in the transcript levels of the genes coding for major receptors and ion channels related to the control of excitability in MNs. The results provide new insight into the mechanisms of MN excitability developing in the subacute postlesion phase below the site of SCT, involving a multiple increase in transcripts coding for AMPAR GluA1 and 5-HTR 1A, 2A, and 2B subunits, and a profound decrease in all GABA_A_R and GlyR subunits, KCC2, KCC3 chloride extruders, and VG SK2 and BK potassium channels. We report that lumbar GL and TA MN groups, despite differential electrical properties [56,57], similarly adapt to a more excitable state interpreted by the transcriptional changes, which may increase the occurrence of extensor and flexor spasms. The alike direction of changes in GL and TA MNs is displayed in Table 1.

### 3.1. Functional Implications of Early Changes in Gene Expression of Glutamatergic Receptors after SCT

Among glutamatergic receptors, ionotropic AMPAR and NMDAR directly affect membrane potentials by controlling the flux of Na^+^ and Ca^2+^ ions. 

AMPAR subunit composition determines their trafficking modes and transmission properties [58,59,60]. The trafficking of AMPARs to and from a synapse is a key mechanism for regulating the strength of synaptic transmission [60,61]. Early studies have shown that the majority of AMPARs in the telencephalon contain the GluA2 subunit [62,63], which dominates major AMPAR transmission properties (single-channel conductance, rectification, and Ca^2+^ permeability) through Arg607, a residue introduced into the GluA2 pore loop by RNA editing at the Q/R site [64]. As already mentioned in the Results, editing at this site is specific for GluA2, as GluA1, GluA3, and GluA4 carry a Gln (Q) at this pore-lining position [65]. What is crucial for the receptor function, a vast majority of GluA2 (>99%) in the normal adult brain is edited to Arg (R), turning AMPAR from Ca^2+^ permeable (CP) to Ca^2+^ impermeable (CI) [65]. Our data which show that in normal rats GluR2 subunit demonstrates the highest transcript level among other AMPAR subunits and that after SCT, GluR2 reveals relative stability of its expression, suggest that also in the spinal cord MNs GluA2 subunit takes a central place in AMPAR composition.

CP-AMPARs that lack the GluA2 subunit have a higher single-channel conductance than the CI-AMPARs which contain GluA2 [66]. For instance, a switch to CP-AMPAR (lacking GluA2 or containing unedited GluA2) increases neuronal excitability in the hypothalamus of hypertensive rats [67], in the hippocampus of mice with seizures [68], in spinal dorsal horn neurons of rats with inflammatory pain [69,70], and also in differentiated human MNs in amyotrophic lateral sclerosis model [71]. 

The Q/R editing of GluA2 may be changed after SCT; this process is primarily regulated by the ADAR2 enzyme, a member of the ADARs family [72,73]. ADAR2 expression changes at some pathologies, thus leading to alterations of Q/R editing of GluA2 [74,75,76]. Therefore we included in our analysis an assay testing transcript level of the *Adarb1* gene, which codes for ADAR2. Our results showed that the mRNA level of *Adarb1* was deceased both in GL and TA MNs after SCT (GL: CN vs. SCT, *p* = 0.02; TA: CN vs. SCT, *p* = 0.01; Mann-Whitney *U*-test, Appendix A), providing a hint that less GluA2 subunits were edited after SCT and a larger influx of Ca^2+^ through AMPAR containing unedited GluA2 can be expected. 

Although in our study we did not characterize the protein levels of AMPAR, we show increased level of transcripts of AMPAR GluA1 and decreased level of GluA3 subunits, with the maintenance of the most abundant transcripts of GluA2 subunit, which suggests that AMPARs containing GluA1 homomers may become more abundant than other subunit combinations in AMPARs produced and undergoing trafficking to synapses in MNs. If so, again, a larger influx of Na^+^ and Ca^2+^ through CP-AMPARs may occur, contributing to increased excitability of MNs.

The transcript level of NMDAR subunits after SCT was decreased. Hypothetically, combined with the increased level of AMPAR GluA1 subunits, it might contribute to “NMDAR to AMPAR functional transition”. This speculation is based on the reasoning that although the single receptor conductance of NMDAR is higher than that of AMPAR [77], NMDARs are blocked by Mg^2+^ at resting conditions and activate slower than AMPARs. Besides, AMPARs represent the majority of Glu receptors on postsynaptic sites [78]. In certain conditions, such as epileptiform activity, CP-AMPARs can be a richer source of intracellular Ca^2+^ than NMDARs, as concluded from the effect of blockade of NMDARs vs. CP-AMPARs; blockade of NMDAR reduced the additional conductance to a lesser extent [79]. Therefore, under the conditions in which the level of NMDAR subunit transcripts is decreased, an increased abundance of CP-AMPARs, if it occurs, might depolarize the postsynaptic membrane by providing a sufficient influx of Ca^2+^. There is experimental evidence to support that reasoning: (1) after NMDAR ablation, the excitability of hippocampal CA3 neurons is enhanced [80]; (2) a significant increase in AMPAR/NMDAR ratio was found in dorsal horn neurons of rats with inflammatory pain which reflects, at least in part, an increase in neuronal excitability [81]. That was associated with a reduction in the quantal amplitude of NMDAR-mediated synaptic currents [69]. Our data suggest that the quantitative switch in Glu receptor expression levels from NMDAR to AMPAR may contribute to an increase or at least a maintenance of MN excitability. How long the increased levels of CP-AMPARs might be maintained is uncertain since, following depolarization, a block of CP-AMPARs by endogenous intracellular polyamines was reported [82]. 

### 3.2. Functional Implications of Downregulation of Gene Expression of GABAergic and Glycinergic Receptors, KCC2 and ion Channels after SCT

We showed that SCT significantly downregulated transcript levels of all examined subunits of GABA_A_R and GlyR in the subacute phase postinjury. In line with our data, decreased gene expression of these receptors was also found in sacral MNs after SCT at sacral segments in rats and was associated with hyperexcitability of these MNs and tail spasticity at the chronic phase postinjury [5]. Importantly, significantly decreased protein levels of GABA_A_R and GlyR on lumbar MNs were reported to contribute to the development of hindlimbs spasticity in rats at 3 weeks after SCT [8]. 

KCC2 is the major chloride (Cl^-^) extruder in mature neurons, sustaining a low-level of intracellular Cl^-^ in basal conditions. By regulating intraneuronal chloride homeostasis, KCC2 has been shown to strongly influence the efficacy and polarity of the GABA_A_R- and GlyR- mediated synaptic transmission [83]. While activated GABA_A_R and GlyR allow an influx of Cl^-^ into neurons to generate an inhibition of neurons by hyperpolarizing the postsynaptic membrane, downregulation of KCC2 protein in MNs was found to weaken postsynaptic inhibition and therefore contribute to spasticity formation after spinal cord injury [23]. Our results in the current study indicated that SCT significantly decreased the transcript levels of the Cl^-^ extruders KCC2 and KCC3, which corresponded with the above observations. Surprisingly, the transcript level of KCC1 was increased after SCT. KCC1 is expressed in the brain, kidney, colon, heart, red blood cells, and dorsal root ganglia [84,85,86,87,88] and is assumed to act as a housekeeping gene but also to mediate the symport of K^+^ and Cl^−^ ions, though a function of that transporter in MNs was not described. The increase of KCC1 may compensate for the profound decrease of KCC2 and KCC3 to protect these MNs.

The transcriptional changes in chloride ion transporters demonstrated in our study suggest that a decreased synthesis of these molecules and subsequent development of reduced inhibition on MNs occur early postinjury, a mechanism described earlier for several populations of spinal neurons and indicated now by our results for both extensor and flexor MNs (Figure 9).

### 3.3. SCT Affects Gene Expression of Ion Channels and Modulatory Receptors

The process of generation of action potential (AP) can be divided into two zones, input (synaptic potentials) and output zone (AP) [89]. Temporal and spatial summation of the postsynaptic potentials (EPSP and IPSP) propagates to the axon hillock and decides whether to fire an AP or not. When a net depolarized membrane potential reaches the threshold, voltage-gated ion channels, such as Na^+^ and Ca^2+^, are responsible for APs generation. Therefore, the expression and abundance of ion channels are crucial for the initiation of APs and the excitability of neurons. 

We have demonstrated that SCT increases the probability of generating EPSPs by upregulating transcript levels of the CP-AMPAR subunits, but reduces the IPSPs probability through downregulating the transcript levels of GABA_A_R and GlyR, possibly decreasing the inhibitory signaling. We reason that after the summation of EPSP and IPSP, an increased net depolarized membrane potential may be expected; in consequence, it may increase the probability of controlling the action of ion channels and generate of APs. 

We show that the changes in gene expression of ion channels after SCT are diverse. The transcript level of two voltage-dependent Ca^2+^ channels, Cav1.3 and Cav2.2, reported to be abundant in the cell body and dendrites of MNs [90], were unaltered, opposite to our expectation, as Cav1.3 was documented as being the major L type Ca^2+^ channel to regulate the persistent inward current in spinal cord MNs and contribute to the formation of spasticity [91,92,93,94]. The major voltage-dependent Na^+^ channel Nav1.6 mRNA in rat spinal MNs was significantly downregulated. In addition, SCT also significantly decreased the mRNA levels of Ca^2+^-activated K^+^ channels SK2 and KCa1.1 in GL and TA MNs. Such patterns of change are difficult to be interpreted. Downregulated Nav1.6 gene may lead to a decrease in the probability of AP generation. Still, on the other side, the decreased SK2 and KCa1.1 might lead to reduced mAHP and enhance the firing of AP and excitability of neurons. Furthermore, in addition to the revealed changes in transcripts, activity and biophysical properties of channels may be modulated by metabotropic receptors. 

Metabotropic mGluRs represent eight subtypes that are classified into three groups [I (mGluR1 and mGluR5), II (mGluR2 and mGluR3), and III (mGluR4, mGluR6, mGluR7 and mGluR8)] based on their physiological activity and receptor structure [95,96]. An early study showed that in the rat lumbar spinal cord mGluR1 and mGluR4 displayed high mRNA expression in the ventral horn MNs, while expression of mGluR3, mGluR5, and mGluR7 mRNA was relatively low, and mGluR2 was not detectable [50]. In addition, Group II and III mGluRs are mainly distributed in the presynaptic membranes [95,96]. In contrast, Group I mGluRs are predominantly located in the postsynaptic membranes [97,98,99,100,101,102], modulating MN firing and the intensity of locomotor-related output [103,104]. Their activation was shown to cause cell depolarization and increased excitability by elevating intracellular Ca^2+^ levels by releasing calcium ions from internal stores or via Ca^2+^ channels [96,105,106,107,108]. Based on the above-mentioned premises, we chose to examine mGluR1 and mGluR5 transcript levels in MNs postinjury; both receptors were shown previously to undergo upregulation in spinal neurons after SCI in rats [109]. In the current study, we identified the same postlesion response of transcripts coding for group I mGluRs. 

In rat superficial spinal dorsal horn neurons, Group I mGluRs have been shown to mediate Ca^2+^ influx via L-type voltage-gated Ca^2+^ channels [110]. Assuming the same mechanism operating in MNs, we may propose that the increased expression, if translated into a changed mGluR protein level, may contribute to the increase of intracellular Ca^2+^ levels in GL and TA MNs.

After SCI, persistent sodium inward currents (Na^+^ PICs) and persistent calcium currents (Ca^2+^ PICs) are spontaneously developed in MNs, contributing to MN firing [18]. The two inward currents are directly generated by voltage-gated Na^+^ channels and Ca^2+^ channels, respectively [32,111,112]. Ca^2+^ channel subtype Cav1.3 was demonstrated to be responsible for generating Ca^2+^ PICs and contributing to the hyperexcitability of MNs after SCI [91,113,114]. However, it is not clear which subtypes of voltage-gated Na^+^ channels mediate the Na^+^ PICs. It is reasonable to propose that Nav1.1 and Nav1.6 may play the roles since they are the most abundant at the transcript level in the two groups of MNs, as revealed in the current study. 

It has been reported that after SCI 5-HTR2 turn into a constitutively active state because their activation does not depend on 5-HT availability in those conditions, signaling through them may also contribute to hyperexcitability of MNs and spasticity [21,22,115]. Furthermore, the generation of Na^+^ PICs and Ca^2+^ PICs is facilitated after the activation of 5-HT_2_ and NAα1 receptors, which modulate the function of the related Na^+^ and Ca^2+^ channels [32,111,112], therefore leading to a general increase of the excitability of MNs. In the current study, in both MN groups, spinalization led to the increased transcript level of 5-HT_1A_ and 5-HT_2B_ receptors, suggesting a positive effect of these receptors’ activation on MN excitability. The increased MN firing frequency may contribute to changes in properties as well, as a result of 5-HT_1A_ inhibition of SK channels and reduced mAHP [116,117]. We conclude that this common response, accompanied by an MN-type specific change (an increase of 5-HT_2A_ in GL MNs and a decrease of 5-HT_2C_ in TA MNs), might be modulated differently in GL and TA MNs.

To conclude, our study shows that the complete transection of the spinal cord leads to fast, adaptive molecular responses of lumbar MNs, which develop in return for the loss of inputs and altered activity of the interneuronal network. These responses are multivariate modulation of transcription of the majority of genes coding for major receptors, transporters, and channels associated with neurotransmission in MNs and are similar in GL and TA MNs. The results suggest that both the ankle extensor and flexor MNs were adapted to be more excitable at the subacute phase postlesion. Moreover, correlation analysis indicated that the set of genes under study undergoes differential regulation in physiological conditions and after injury. 

Our work provided an overview of the changes which form the landscape of transcriptional alterations of genes coding for the proteins operating at the membrane and may serve to help understand the molecular background of the pathology of SCI and SCI-induced spasticity. The characterization of the protein changes in parallel with electrophysiological studies is needed to understand the impact of detected changes on MN function. Moreover, this study and our ongoing studies show that despite the shift priming increased excitability after SCT, these MNs do not display abnormal morphology and can respond to BDNF overexpression induced in the lumbar segments that improve locomotor abilities short-term. Disclosing the molecular targets which are preferentially altered in MNs in response to increased BDNF levels may add to further understanding of factors decisive for early functional improvement but developing pathology long-term.

## 4. Materials and Methods

### 4.1. Animals

Eighteen adult male Wistar rats (260–340 g) were used in this experiment. Rats were randomly divided into control group (CN; N = 9) and the spinal cord transection group (SCT; N = 9). Experimental protocols involving animals, their surgery, and care complied with the guidelines described in the Institutional Review Board Statement. Rats used in the experiments were outbred colonies purchased from the Medical University of Białystok (Białystok, Poland). During the experiments, rats were housed in the Animal Facility of the Nencki Institute of Experimental Biology (Warsaw, Poland), under standard humidity and temperature conditions, at 12 h light/dark cycle, with free access to water and pellet food. Rats were housed in groups of 4–6 prior to surgery and individually after surgery. All efforts were made to minimize the number of animals used and their suffering.

### 4.2. Retrograde labeling of MNs 

To identify MNs, all the rats were injected with cholera toxin subunit B conjugated with Alexa Fluor™ 594 or Alexa Fluor™ 488 (20 µL, 0.1 mg/mL working solution in phosphate-buffered saline, Molecular Probes, US) into GL and TA muscles, respectively, 2 weeks before transection of the spinal cord (Figure 1A,B). The procedure of neuronal tracing was as described in [37]. Briefly, the rats were given a subcutaneous injection of butorphanol analgesic (Butomidor, Richter Pharma, Wels, Austria; 3.3 mg/kg) as a premedication and then were anesthetized with isoflurane (Baxter, Lessines, Belgium, 1–2.5% in oxygen) via facemask. The skin of the hindlimb was shaved, disinfected with 3% hydrogen peroxide, and cut. The biceps femoris muscle was incised to expose the underlying GL and TA muscles. The tracers were injected bilaterally using a Hamilton microsyringe with a 22-gauge needle attached. To avoid leakage and degradation of the tracers, the needle was kept in the muscle body for 5 min. The injection site was cleaned carefully with warm 0.9% NaCl after retraction of the needle, and the skin was closed with surgical sutures. After the surgery, the rats were placed in warm cages and inspected until they fully awakened. Plastic collars (Harvard Apparatus, Holliston, MA, USA) were put on each animal during the first postinjury day to protect their wounds from licking, and rats were returned to individual cages with access to food and water. Antibiotic Baytril (Enrofloxacinum 5 mg/kg; Bayer GmbH, Leverkusen, Germany) and analgesic Tolfedine (Tolfenamic acid 4%, 4 mg/kg, s.c., Vetoquinol S.A., Lure Cedex, France) were administered daily over the first five postoperative days to prevent infection.

### 4.3. Complete Spinal Cord Transection

Nine rats were spinalized at a low thoracic level of the spinal cord two weeks after retrograde labeling of MNs. Briefly, the rats were anesthetized as described in the procedure preceding the MN tracing. The skin on the back was opened, and the muscles around were removed with a scalpel to expose the low thoracic vertebrae. A laminectomy was performed at the level of T9/10 vertebrae. The dura mater was opened, and *lignocainum hydrochloricum* (2% solution; Polfa Warszawa S.A., Poland) was applied to the surface of the spinal cord, which was then completely transected using surgical scissors. The gap between the rostral and caudal ends was enlarged by aspiration to about 1 mm, washed with warm 0.9% NaCl and dried with absorbable cellulose. After careful inspection of the lesion area, the surrounding tissues were closed with surgical sutures, and the edges of the cut skin were joined with stainless steel staples. After the surgery, the antibiotic Sultridin (24% Sulfadiazine + Trimethoprim, 30 mg/kg, Norbrook, Ireland) was administered over five days, and the analgesic Vetaflunix (Flunixin meglumine, 2.5 mg/kg, VET AGRO, Poland) for three days postlesion. The bladder was manually emptied twice daily, and their bodies were cleaned if necessary.

### 4.4. Tissue Preparation and Laser Microdissection (LMD)

Rats were deeply anesthetized with a lethal dose of Morbital (pentobarbital 120 mg/kg, Biowet Puławy Ltd., Poland) and transcardially perfused with 250 mL ice-cold 0.01 M PBS ([in mM] 154 NaCl, 1.3 Na_2_HPO_4_, 2.5 NaH_2_PO_4_, pH 7.4). The vertebral column was excised and placed on ice. The spinal cord lumbar (L) 3–6 segments were rapidly dissected in a cold room and were immediately put in dry ice-precooled tubes and stored at −80 °C until further sectioning. For sectioning, L3–6 segments were surrounded by Jung tissue-freezing medium (Leica, Nussloch, Germany, cat no 14020108926) and cut into 25 µm thick longitudinal sections on the cryostat (Leica CM1850) at −20 °C. The sections were mounted onto RNase-free PET-Membrane Frame Slides (Leica, No. 11505190), 4 sections/slide. Slides were put immediately into the dry ice precooled box and stored at −80 °C until processing for LMD. 

Prior to LMD, slides were dehydrated with ethanol in the following order: 70%, 80%, 90%, and 100% ethanol, 30 s each, followed by 100% ethanol (fresh) for 30 s. After dehydration, sections were cleared in 100% xylene washes (1× for 30 s, 1× for 180 s). Next, slides were air dried for 3–5 min. Leica Laser Microdissection system LMD7000 was used to isolate the MNs. The RNase-free caps of 0.2 mL tubes were put on the tube holder for sample collection and filled with 20 ul extraction buffer (from Arcturus™ PicoPure™ RNA Isolation Kit, KIT0204, Applied Biosystems, Thermo Fisher Scientific, Waltham, MA, USA). The air-dried slide was placed on the slide holder of the LMD 7000 system. Under the LMD system microscope (objectives × 10, 0.32 NA and × 63, 0.7 NA) attached to the software unit, MNs were identified, selected, and then cut with a UV laser (Figure 1B). After the collection was complete for one slide, the microfuge tube was removed from the holder. Collected MNs were lysed by incubating the sample in the buffer for 30 min at 42 °C. The lysates were spun down for 2 min at 800g to collect them at the bottom of the tubes. The sample tubes were immediately frozen by placing them on dry ice and then stored at −80 °C. To limit RNA degradation, samples were collected for up to 60 min per slide. 

### 4.5. RNA Isolation and qPCR

The total RNA from the GL and TA MNs was extracted using Arcturus PicoPure™ RNA Isolation Kit (No. KIT0204, Thermo Fisher Scientific, Waltham, MA, USA) according to the manufacturer’s instructions. RNA was then pre-amplified and reverse-transcribed using a QuantiTect^®^ Whole Transcriptome kit (No.207043, 207045, Qiagen, Hilden, Germany). The final concentration of the cDNA was determined by utilizing Quant-iT™ PicoGreen™ dsDNA Assay Kit (No. P7589, Thermo Fisher Scientific, Waltham, MA, USA). The qPCR was used to analyze gene expression levels with glyceraldehyde-3-phosphate dehydrogenase (*Gapdh*) as an internal control gene. The qPCR reactions were performed in a 20 µL reaction mixture containing 7.4 μL PCR grade H_2_O, 10 μL LightCycler 480 Probes Master solution (Roche, cat no. 04887301001), 0.4 µL forward primer (20 µM) of target gene, 0.4 µL reverse primer (20 µM) of target gene, 0.4 µL probe (10 µM) of target gene, 0.1 µL forward primer (20 µM) of *Gapdh* gene, 0.1 µL reverse primer (20 µM) of *Gapdh* gene, 0.2 µL probe (10 µM) of *Gapdh* gene, 1 μL diluted cDNA sample. Probes and primers are listed in Appendix A. Roche LightCycler 96 was used to run the reactions with the following thermal cycling profile: preincubation at 95 °C for 10 min, 65 cycles of denaturation at 95 °C for 10 s, annealing at 60 °C for 10 s, and extension at 72 °C for 10 s. Data were analyzed by using the LightCycler software. The 2^−ΔΔCt^ method was used for relative quantification of gene expression level based on the target and reference genes’ Ct (cycle threshold) value, the cycle number at which the amplification curve reaches the threshold line.

### 4.6. Statistical Analysis

Data were analyzed by the STATISTICA 13.1 software (StatSoft Inc, Tulsa, OK, USA). Shapiro-Wilk test was used to verify the normality of distribution of all data. Mann-Whitney U-test was used to compare two independent groups of samples throughout the study because the normality of data distribution was not met in the case of some data. The Wilcoxon test was used for dependent samples. The scatter plot and bar graphs were drawn using GraphPad Prism 9.2.0 software (trial version, GraphPad Software, San Diego, CA, USA). Pearson’s correlation was calculated based on ΔCt by corr.test library in R and visualized by corrplot library.

## Figures and Tables

**Figure 1 ijms-23-11133-f001:**
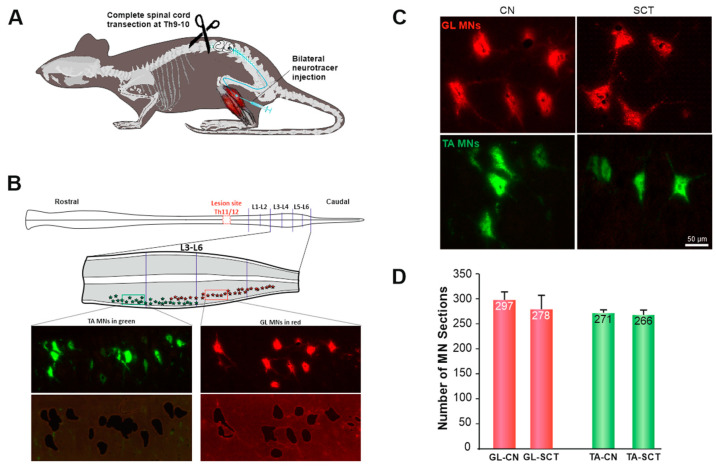
Experimental procedures (**A**,**B**), morphology (**C**), and number (**D**) of GL and TA MN sections isolated from the intact (CN) and spinalized (SCT) rats. A: The GL and TA MNs in the spinal cord were traced with intramuscularly injected Alexa Fluor 555- (GL) and Alexa Fluor 488 (TA) conjugated cholera toxin subunit B (CT-B). Two weeks later, the rats received a complete transection of the spinal cord at the thoracic T9–10 vertebrae level. B: Schematic of the spinal cord with the lesion site (marked in red) at the T11/12 segments and the lumbar segments under study; the distribution of GL MNs (red) and TA MNs (green) in L3–6 is shown on the diagram enlargement (middle panel). Lower panel: representative photomicrographs showing GL and TA MNs before laser microdissection and the tissue section remaining after microdissection. The traced neurons are also shown enlarged in Figure 1C, documenting that the morphology and size of MNs are comparable between groups. D: the number of GL and TA MN sections isolated from the CN and SCT groups of rats. Assuming the diameter of rat lumbar MNs to be in a range of ~20–80 µm [42], two or three MN sections (thickness 25 µm) have been estimated to reconstruct one MN. Data are means +/− SEM. Nine rats per group were examined. Figure 1B has been modified from our study [27].

**Figure 2 ijms-23-11133-f002:**
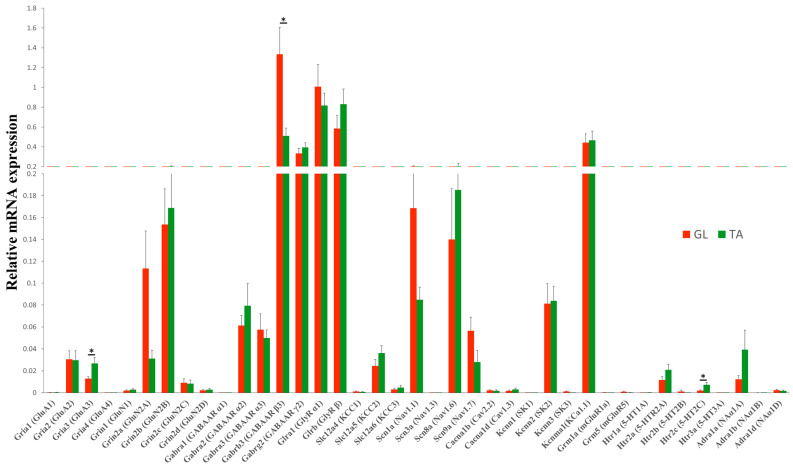
Steady-state mRNA abundance of receptor and ion channel subunits in GL and TA MNs of the intact rats. Comparison of transcript levels of genes encoding major subunits of ionotropic excitatory (AMPA, NMDA), inhibitory (GABA_A_, Gly), modulatory (mGlu, 5-HT, and NA) receptors, ion channels (Na^+^, Ca^2+,^ and K^+^) and K^+^-Cl^−^ cotransporters KCCs shows that all groups of genes showed similar expression pattern in GL and TA MNs. N = 9 rats. Data are means +/− SEM. Wilcoxon test, * *p* < 0.05.

**Figure 3 ijms-23-11133-f003:**
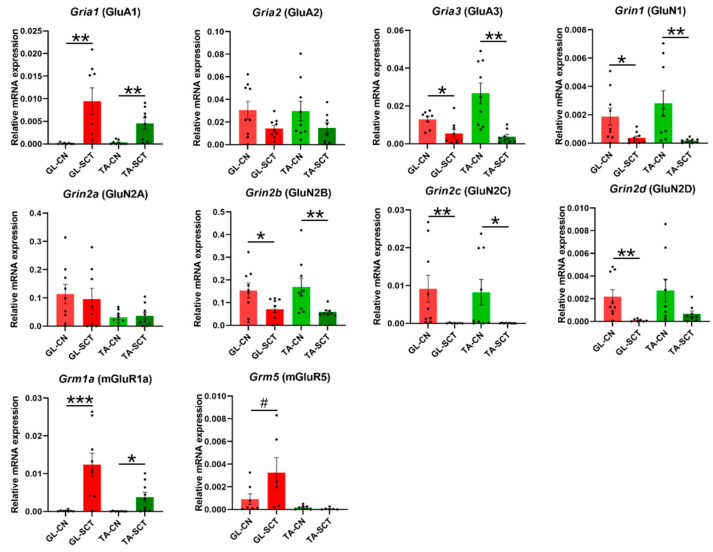
SCT differentially affects the transcript levels of AMPAR, NMDAR, and mGluRs subunits, with the same impact on GL and TA MNs. SCT led to a significant increase in the *Gria1* (GluA1) subunit and a decrease in the *Gria3* (GluA3) subunit. A significant decrease in transcripts of all subunits of NMDAR except for *Grin2a* (GluN2A) was found. Upregulation of the transcript level of *Grm1a* (mGluR1A) in GL and TA MNs was accompanied by an increase in the transcript level of *Grm5* (mGluR5) in GL MNs but not in TA MNs. Data are means +/− SEM. Mann-Whitney *U*-test, # *p* = 0.074; * *p* < 0.05; ** *p* < 0.01; *** *p* < 0.001.

**Figure 4 ijms-23-11133-f004:**
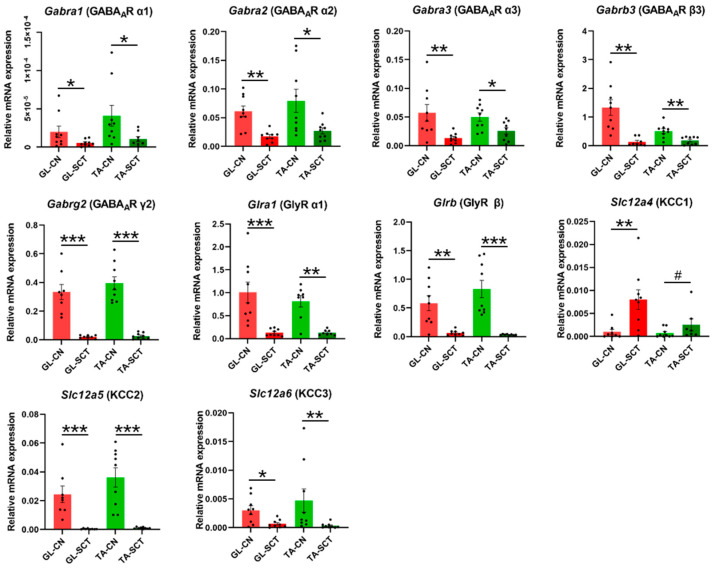
SCT downregulates the transcript level of all subunits of GABA_A_ and Gly receptors, as well as KCC2 and KCC3 transporters, with the same impact on GL and TA MNs. The opposite response was found for the KCC1 transporter, both in GL and TA MNs. Data are means +/− SEM. Mann-Whitney *U*-test, # *p* = 0.072; * *p* < 0.05; ** *p* < 0.01; *** *p* < 0.001.

**Figure 5 ijms-23-11133-f005:**
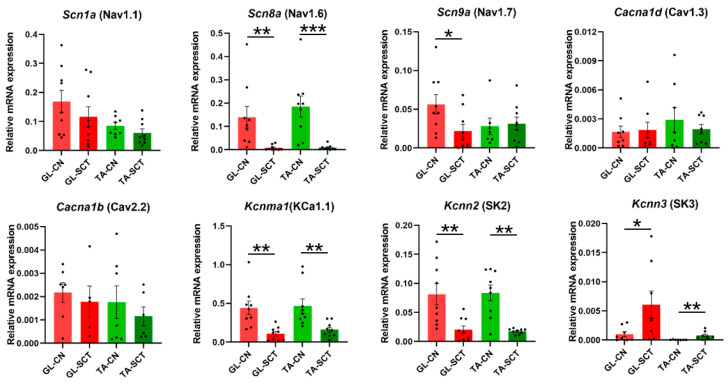
SCT downregulates the transcript level of the Nav1.6, KCa1.1 (BK), and SK2 channels similarly in GL and TA MNs. Expression of voltage-dependent Ca^2+^ channels *Cacna1d* (Cav1.3) and *Cacna1b* (Cav2.2) was unaffected by SCT. Data are means +/− SEM. Mann-Whitney *U*-test, * *p* < 0.05; ** *p* < 0.01; *** *p* < 0.001.

**Figure 6 ijms-23-11133-f006:**
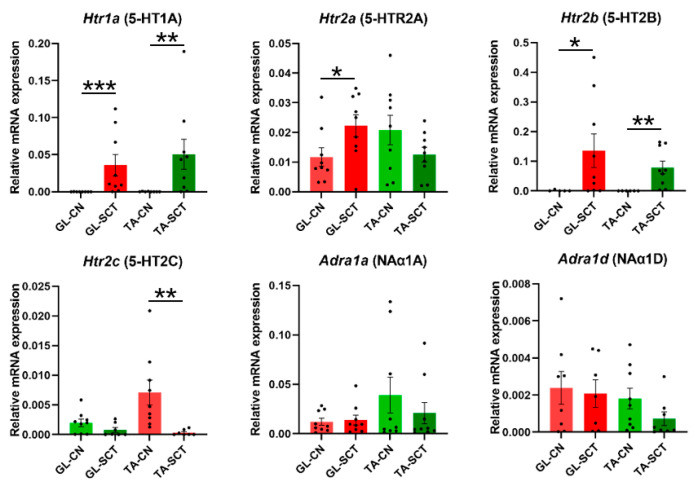
SCT upregulates the transcript level of 5-HT 1A and 2B receptors similarly in GL and TA MNs. Increased transcript level of *Htr1a* (5-HT1A), *Htr2a* (5-HT2A), and *Htr2b* (5-HTR2B) subunits of 5-HT receptors in GL and TA MNs is accompanied by a decrease in *Htr2c* (5-HT2C) subunit in TA MNs but not GL MNs. SCT does not alter gene expression of NA receptor subunits. Data are means +/− SEM. Mann-Whitney *U*-test, * *p* < 0.05; ** *p* < 0.01; *** *p* < 0.001.

**Figure 7 ijms-23-11133-f007:**
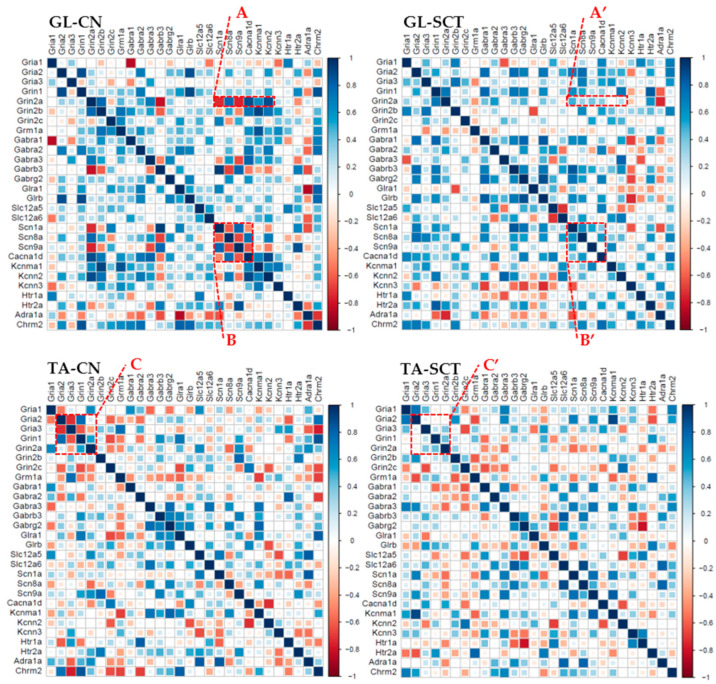
Changes in correlated gene expression as a result of SCT in GL and TA MNs. The order of the genes along the X- and Y-axis of each correlogram is the same for comparing the changes between groups.

**Figure 8 ijms-23-11133-f008:**
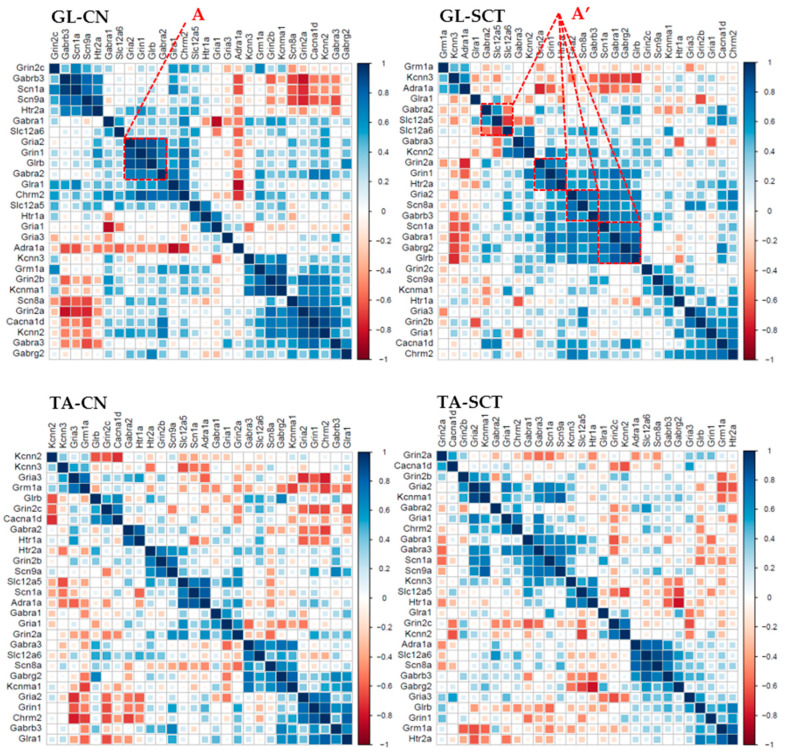
Changes in clustered correlated gene expression as a result of SCI in GL and TA MNs. The order of the genes along the X- and Y-axis of each correlogram are automatically clustered depending on the R-values of the genes.

**Figure 9 ijms-23-11133-f009:**
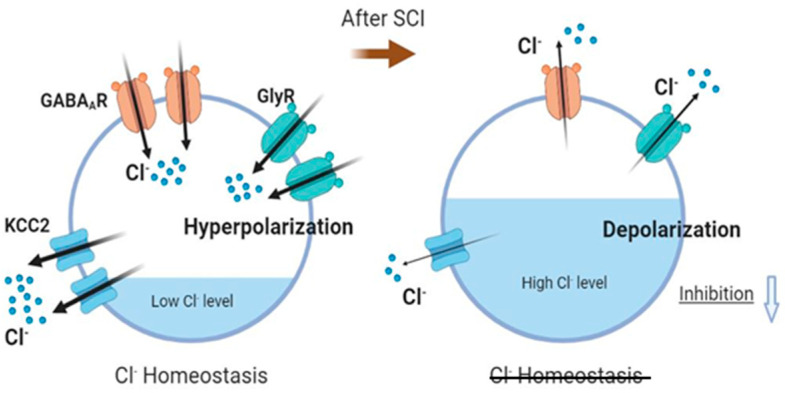
Downregulation of inhibitory receptors and chloride ions extruder KCC2 in MNs occurs early postinjury period causing a decrease of postsynaptic inhibition. In basal conditions, KCC2 maintains a low-level of intracellular chloride ions, and the activated GABA_A_Rs and GlyRs allow an influx of chloride ions, generating a hyperpolarization. After SCT, a decrease of the KCC2 may lead to the accumulation of high concentrations of Cl^−^ inside MNs, and evoke a depolarizing instead of an inhibitory response, suggesting the possibility of a decrease in inhibition. Illustration created with BioRender.com, 1 June 2022.

**Table 1 ijms-23-11133-t001:** Direction of changes in gene expression in GL and TA MNs at 2 weeks after SCT.

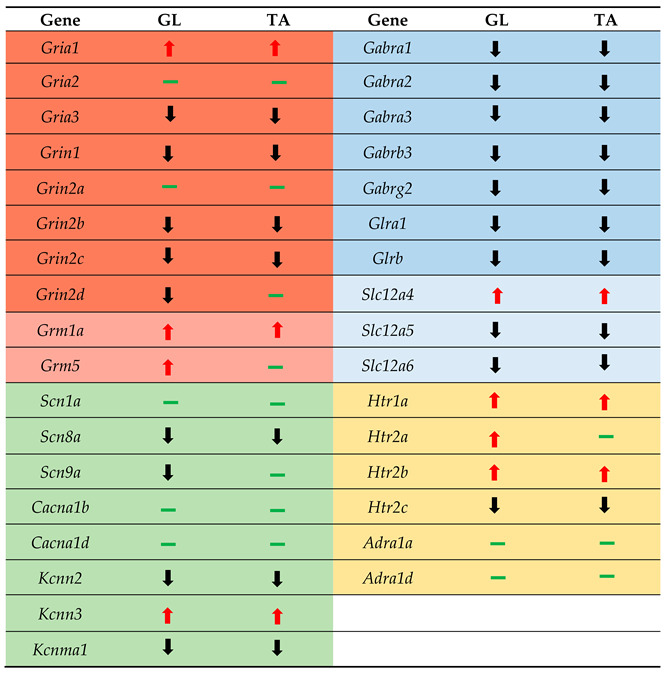

## Data Availability

Data available upon request.

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
