# Peer review of "Molecular Identification of Pro-Excitogenic Receptor and Channel Phenotypes of the Deafferented Lumbar Motoneurons in the Early Phase after SCT in Rats"

_ijms, 2022, doi:10.3390/ijms231911133_

Round 1
Reviewer 1 Report
1. Authors may consider combining Figure 1 and 10.
2. It is preferred that authors provide a functional characterization of reported altered expression. Subtle changes to MN function are expected and which maybe directly relevant for the claims associated to physiology of SCI induced spasticity.
3. Authors are advised to provide a suitable functional and microscopic evidences for the changed receptor/channel expression.
4. What happens to the synapses e.g. number of synaptic sites in response to SCT? Is reported altered expression is similarly changed on synaptic sites? Picture quality can be further improved.
5. Please revisit the manuscript for format related, grammatical or typos related issues.
Author Response
Answers (in Red) to the reviewer reports:
Open review 1:
Comments and Suggestions for Authors
- Authors may consider combining Figure 1 and Figure 10.
RESPONSE 1.1. Thank you for the reasonable suggestion. We constructed a new Figure 1 by combining the two figures.
- It is preferred that the authors provide a functional characterization of the reported altered expression. Subtle changes to MN function are expected and which maybe directly relevant for the claims associated to physiology of SCI induced spasticity.
- Authors are advised to provide a suitable functional and microscopic evidences for the changed receptor/channel expression.
RESPONSE 1.2 for point 2 and 3. Thank you very much for the valuable advice. We fully agree that functional characterization of altered expression of the reported genes at the protein level and electrophysiology study would provide more direct evidence to reveal changed excitability of MNs. While we share the view that both aspects cannot be overestimated, the experiments which would be required to reach the goal are beyond our current possibilities.
The material was collected to answer the question on regulation of function of specific, unequivocally identified motoneuron types at the level of expression of the genes; thus it served highly sensitive RT-PCR assays on LMD-isolated, single neurons, making it possible to obtain a selective and specific response on the same material for comparisons of the transcript levels, what is rare in the literature of the subject. Technical processing of the tissue does not allow to continue with this material for any other assay. Also, acute electrophysiological experiments would have to be done on the other group of animals than those intended for protein assay to obtain reliable data. For both purposes new animal experiments would have to be done and still no direct answer would be obtained, as we would deal with separate animals and motoneurons in each experimental group. That project would take over a year to conduct and analyze, delaying the presentation of the current output.
We think that in the submitted manuscript, the asset is a delivery of an overview of the transcriptional changes of thoughtfully selected genes which relate to the excitability of MNs. We expect that disclosure of these transcriptional changes increase our understanding of pathology of SCI and SCI- induced spasticity.
- What happens to the synapses e.g. number of the synaptic sites in response to SCT? Is altered expression is similarly changed on synaptic sites?
RESPONSE 1.3. Our previous studies showed that SCT leads to a long term decrease (5-6 weeks) in markers of synaptic nerve endings abutting motoneurons ([1] (the degree of loss – 18% of synaptophysin labeling surrounding large neurons of the ventral horn in lumbar L3/4 segments; -26% in synaptic zinc staining), a population of cholinergic [2] and excitatory (Glutamatergic) as well as inhibitory (GABAergic) terminals [3, 4]. These responses of inputs to the injury prompted us to ask whether postsynaptic receptors adapt to developing changes. In the submitted manuscript, our results showed decreased transcript level of inhibitory GABAergic and glycinergic receptors which is in line with reduced inhibitory (GABAergic) inputs and suggests that the compensatory mechanism of up-regulation is not activated. The changes in the transcript levels of glutamatergic receptors were diverse, with increased AMPAR GluA1 and mGluR1, and decreased NMDAR.
- Please revisit the manuscript for format related, grammatical or typos related issues.
RESPONSE 1.4. According to your suggestion, we revisited the manuscript and corrected issues which we traced.
The references quoted in the responses:
- Macias, M.; Nowicka, D.; Czupryn, A.; Sulejczak, D.; Skup, M.; Skangiel-Kramska, J.; Czarkowska-Bauch, J., Exercise-induced motor improvement after complete spinal cord transection and its relation to expression of brain-derived neurotrophic factor and presynaptic markers. BMC Neurosci 2009, 10, 144.
- Skup, M.; Gajewska-Wozniak, O.; Grygielewicz, P.; Mankovskaya, T.; Czarkowska-Bauch, J., Different effects of spinalization and locomotor training of spinal animals on cholinergic innervation of the soleus and tibialis anterior motoneurons. Eur J Neurosci 2012, 36, (5), 2679-88.
- ZiemliÅ„ska, E.; Kügler, S.; Schachner, M.; Wewiór, I.; Czarkowska-Bauch, J.; Skup, M., Overexpression of BDNF Increases Excitability of the Lumbar Spinal Network and Leads to Robust Early Locomotor Recovery in Completely Spinalized Rats. PLoS ONE 2014, 9, (2).
- Grycz, K.; Glowacka, A.; Ji, B.; Czarkowska-Bauch, J.; Gajewska-Wozniak, O.; Skup, M., Early pre- and postsynaptic decrease in glutamatergic and cholinergic signaling after spinalization is not modified when stimulating proprioceptive input to the ankle extensor alpha-motoneurons: Anatomical and neurochemical study. PLoS One 2019, 14, (9), e0222849.

Reviewer 2 Report
This manuscript does a very good job providing a new insight into the molecular mechanisms of MN excitability developing in the subacute phase below the site of SCT, involving a multiple increase in the transcripts coding for AMPAR and 5HTR subunits, and a profound decrease in GABAaR, GlyR subunits, and KCC2. Altogether demonstrating that both MNs groups adapt to a more excitable state, increasing the occurrence of extensor and flexor spasms.
The paper is very well written, and does a perfect job walking you through the background with a good introduction, and then to the well explained discussion of the data.
The amount of data presented is huge, with 10 figures, some of them including 10 graphs, and the authors made a good job organizing both the graphs and the written results to make the results and their conclusions accessible to the readers.
I only have a minor comment about the graphs, as it is a bit annoying to find graphs in the same figure with such scale differences of the axes, although I also understand the nature of the results presented, and why the axes have such differences in the scales.
Author Response
Answers (in Red) to the reviewer reports:
Open review 2:
Comments and Suggestions for Authors
This manuscript does a very good job providing a new insight into the molecular mechanisms of MN excitability developing in the subacute phase below the site of SCT, involving a multiple increase in the transcripts coding for AMPAR and 5HTR subunits, and a profound decrease in GABAaR, GlyR subunits, and KCC2. Altogether demonstrating that both MNs groups adapt to a more excitable state, increasing the occurrence of extensor and flexor spasms.
The paper is very well written, and does a perfect job walking you through the background with a good introduction, and then to the well explained discussion of the data.
The amount of data presented is huge, with 10 figures, some of them including 10 graphs, and the authors made a good job organizing both the graphs and the written results to make the results and their conclusions accessible to the readers.
I only have a minor comment about the graphs, as it is a bit annoying to find graphs in the same figure with such scale differences of the axes, although I also understand the nature of the results presented, and why the axes have such differences in the scales.
RESPONSE 2.1. Thank you very much for your positive evaluation of the work and empowering comments.
Regarding the graphs, we decided to reduce their number from 10 to 9, by combining Fig 1 with Fig 10, as suggested by one of the reviewers. In subsequent Figures, the differences between the scales of the Y axes are dependent, as you have noticed, on the expression levels of different genes. In a process of preparation of the manuscript we considered to unify the scale for all the genes. However, because some genes have much higher expression level than other genes, by using one scale we would make the genes with lower expression invisible in the graph. While considering the use of a fold-change unit we waived that decision, to maintain the information on transcript levels in control animals. Hope this explanation justifies the selection of the scale.

Reviewer 3 Report
This submission reported that complete SCT at a low thoracic level leads to early, significant changes in expression of the genes coding for central receptors and ion channels related to the control of excitability in motoneurons. This study was well designed, data are sound. However, there are still several issues that need to be imported. A revision is suggested.
1. English editing is suggested.
2. Please reduce keywords.
3. Please provide scale bars of Fig. 1
4. Protein expression levels are also needed to study (critical molecular, instead of just conducting mRNA assay).
5. Please discuss the limitation and clinical implications of this study.
Author Response
Answers (in Red) to the reviewer reports:
Open review 3:
Comments and Suggestions for Authors
This submission reported that complete SCT at a low thoracic level leads to early, significant changes in expression of the genes coding for central receptors and ion channels related to the control of excitability in motoneurons. This study was well designed, data are sound. However, there are still several issues that need to be imported. A revision is suggested.
- English editing is suggested.
RESPONSE 3.1. The manuscript was revisited and English editing was done.
- Please reduce keywords.
RESPONSE 3.2. We reduced the number of keywords to 9.
- Please provide scale bars of Fig. 1
RESPONSE 3.3. We added the scale bar in the new Figure 1.
- Protein expression levels are also needed to study (critical molecular, instead of just conducting mRNA assay).
RESPONSE 3.4. We share your opinion on the importance of molecular study at the level of protein expression. We admit, that expanding the study to determine protein responses would be an important step, but the experiments required to obtain data on the protein are beyond our current possibilities. A real difficulty is to operate with the reliable antibodies to investigate distribution and level of the proteins under study on tissue sections; we approached that with a minor success, but will continue a search and trials. On the other hand ELISA assays are still not sensitive enough to make an assay with the use of small protein samples obtained from the collection of GL and TA motoneurons, to get data characterizing changes in the identified groups of motoneurons.
For the purpose of the current study the material was collected to answer the question on regulation of function of specific, unequivocally identified motoneuron types at the level of expression of the genes; thus it served highly sensitive RT-PCR assays on LMD-isolated, single neurons, making it possible to obtain a selective and specific response on the same material for comparisons, what is rare in the literature of the subject and provides a direct information on the motoneuron states.
For a purpose of protein analysis by in situ immunofluorescence, new animal experiments would have to be done and still no direct answer would be obtained, as we would deal with separate animals and motoneurons in each experimental group. That project would take over a year to conduct and analyze data, delaying the presentation of the current output. I hope our response will meet with the Reviewer understanding.
We think that in the submitted manuscript, the asset is a delivery of an overview of the transcriptional changes of thoughtfully selected genes which relate to the excitability of MNs. We expect that the landscape of transcriptional changes of the proteins operating at the membrane provides meaningful and useful information on the pathology of SCI and SCI- induced spasticity.
- Please discuss the limitation and clinical implications of this study.
RESPONSE 3.5. Thank you for the suggestion. We added more information regarding the two points in the Discussion part.
Reviewer 4 Report
The manuscript submitted by Benjun Ji and colleagues demonstrates the changes in expression of a set of genes in a model of spinal cord injury. The study represents a screening research. In my opinion, two major shortcomings can be pointed out.
1. Poor rationale of the selected genes in Introduction or Results section.
For instance, there are numerous subtypes of voltage-gated calcium channels that significantly contribute to AP and burst activity generation/spreading in neurons. These channels were found at presynaptic or postsynaptic terminals. However, the authors have chosen only Cav1.3 and Cav2.2 channels. Why? The similar trend is observed in the case of 5-HT receptors which are subdivided into 7 families and numerous subfamilies and in the case of mGluRs (the authors study only Group I receptors, but what about Group II and III?).
2. Most of the speculations in Discussion seem as conclusions. However, these conclusions are not supported by experimental data and cannot be drawn from PCR results. For instance, the conclusions about EPSP/IPSP generation or Cl- homeostasis cannot be drawn only from PCR data. Thus, Discussion must be completely revised.
Minor points
"and no labeled apoptotic profiles". How can the neurons be attributed to apoptotic using only tracers but not specific fluorescent dyes (Hoechst 33342, NucView 488, Annexin V etc.)?
-I would recommend reorganizing some figures. The authors compare GL and TL and present two separate panels. It would be better combining these panels. Please, also consider the combining of panels in Fig.3 to single panel (similar for Figures 4 and 5).
-Please, add the scale bar in Figure 1A and clearly indicate what green and red color mean. Why does the background in green channel (bottom line, left image) is red?
-The authors indicate that AMPARs Ca2+-permeability is determined by subunit composition and describe the role of Q/R editing in AMPAR-mediated Ca2+-inflow tuning. It is known that almost 100% of GluA2 pre-mRNA is edited in adult animals. However, ADARs expression can be changed at some pathologies, thus leading to alterations in GluA2 editing. Please, discuss it.
Author Response
Answers (in Red) to the reviewer reports:
Open review 4:
Comments and Suggestions for Authors
The manuscript submitted by Benjun Ji and colleagues demonstrates the changes in expression of a set of genes in a model of spinal cord injury. The study represents a screening research. In my opinion, two major shortcomings can be pointed out.
- Poor rationale of the selected genes in the Introduction or Results section.
For instance, there are numerous subtypes of voltage-gated calcium channels that significantly contribute to AP and burst activity generation/ spreading in neurons. These channels were found at presynaptic or postsynaptic terminals. However, the authors have chosen only Cav1.3 and Cav2.2 channels. Why? The similar trend is observed in the case of 5-HT receptors which are subdivided into 7 families and numerous subfamilies and in the case of mGluRs (the authors study only Group I receptors, but what about Group II and III?).
RESPONSE 4.1. Thank you very much for this comment and questions which allow us to justify the choice and show that it is not random.
We definitely agree with the Reviewer, that there are many other molecules which are critical for the properties of the neurons in general. However, in our study there is a limitation of the material availability, because it is acquired from the selected, clearly identified rat spinal motoneurons, a specific target of our search. cDNA obtained from the mRNA samples, even if acquired with optimized methodology, does not allow to examine all the genes. Therefore, we had to thoroughly choose representative ones, which have been well documented in the literature to be important for the excitability of the MNs, or we had reason to believe they could be important to this feature. Here we provide our reasoning, which concerns the receptors you are asking about:
Ca2+ channels: Immunocytochemistry study [1] showed that in the rat spinal cord ventral horn, P/Q-type Cav2.1 is primarily localized in nerve terminals. On the contrary, N-type Cav2.2 is localized to the cell bodies, dendrites and nerve terminals of motoneurons. L-type Cav1.3 is also localized in motoneurons throughout the ventral horn. R-type Cav2.3 was also found in the cell body of motoneurons. Among them, Cav1.3 has been well documented to be the major L type Ca2+channel to regulate the persistent inward current in spinal cord MNs and contribute to development of spasticity [2-5]. Cav2.2 is expressed not only in motoneurons, but also widely expressed in the dorsal horn neurons carrying nociceptive information in the spinal cord. Therefore, we considered that Cav2.2 might be important in the sensorimotor circuit in the spinal cord. In sum, based on their subcellular distribution and functions, we chose Cav1.3 and Cav2.2 as strong candidates that may contribute to the excitability of motor neurons after SCT.
mGluRs: Early study showed that, in the rat lumbar spinal cord, mGluR1 and mGluR4 displayed high mRNA expression in the ventral horn motoneurons, while expression of mGluR3, mGluR5 and mGluR7 mRNA was relatively low and mGluR2 was not detectable [6]. In addition, Group I mGluRs have been suggested to be mainly located in the postsynaptic membrane, while Group II and III mGluRs are mainly presynaptic [7-9]. Therefore, based on the above-mentioned premises, we chose to examine Group I mGluR which can directly contribute to a number of intracellular events to regulate excitability of MNs.
5-HT receptors: According to the published reports [10, 11], 5-HTR1 (A, Ð’, D), 5-HTR2 (A, B, C), 5-HTR3 (low expression, data was not shown in our study), 5-HTR5A and 5-HTR7 are expressed in spinal cord MNs. In our study, we focused on 5-HTR2, because activation of that receptor type on MNs have been well shown to facilitate generation of voltage dependent Ca2+ and Na+ persistent inward currents (PICs) through modifying the behavior of the Ca2+ channels (low-threshold L-type calcium channels CaV1.3) and Na+ channels via intracellular second messenger systems [12-16]. PICs increase MN excitability through amplifying subsequent synaptic inputs and generating plateau potentials on MNs. Moreover, it has been reported that after SCI 5-HTR2 turn into a constitutively active state; because their activation does not depend on 5-HT availability, signaling through them may contribute to hyperexcitability of MNs and spasticity [17-19]. The observation of the constitutive active state raised the question on the mechanism of 5-HTR2 regulation at the transcript level transcript; in our opinion that was a strong rationale to choose these receptors for the study.
- Most of the speculations in Discussion seem as conclusions. However, these conclusions are not supported by experimental data and cannot be drawn from PCR results. For instance, the conclusions about EPSP/IPSP generation or Cl-homeostasis cannot be drawn only from PCR data. Thus, Discussion must be completely revised.
RESPONSE 4.2. Thank you for this comment. As we described earlier, we focused on the transcriptional regulation of the two types of MNs. Following an overview of the transcriptional changes of selected genes which relate to the excitability of MNs, in the Discussion we speculated on the probable protein alterations and their functional consequences to extend the perspective. According to your suggestions, we revised the Discussion to make it less speculative and up to the point.
Minor points
"and no labeled apoptotic profiles". How can the neurons be attributed to apoptotic using only tracers but not specific fluorescent dyes (Hoechst 33342, NucView 488, Annexin V etc.)?
RESPONSE 4.3. Thank you for that comment that drew our attention to the necessity of carrying on additional labeling of cells and documenting morphological features of neurons and accompanying cells. We conducted staining with Hoechst 33342 on the spinal cord transverse sections which contained traced (Cholera toxin) GL and TA MNs from control (CN) and spinalized (SCT) animals, and analyzed ventral horn regions in sampled tissue. In effect neither chromatin condensation nor apoptotic profiles of motoneurons were detected. We decided to add the images to the manuscript and prepared a Figure (shown here) which is proposed to be included in the manuscript as a Supplementary Figure A (Figure S1A).
In addition, because Hoechst staining is known to provide a weak signal from motoneuron nuclei owing to predominance of euchromatin in the physiological state of the neuron (in the Figure the sections were exposed long enough to show clearly MN nuclei; in these conditions nuclei of glial cells including oligodendrocytes which are recognizable by the dense chromatin adjacent to the nuclear pores are overexposed), we performed additionally Nissl staining. We used NeuroTrace 435/455 blue-fluorescent Nissl stain (ThermoFisher Scientific) which is selective for the Nissl substance characteristic of neurons and provides more sensitivity than traditional histological dyes like toluidine blue or cresyl violet. We took advantage of the property of the Nissl substance which redistributes within the cell body in injured or regenerating neurons, providing a marker for the physiological state of the neuron. The representative images are shown on the attached figure below and will be a supplementary Figure S1B. As shown the distribution of the Nissl substance in the MNs and cell morphology are comparable between CN and SCT animals. Based on this labeling we concluded that neither GL and TA, nor other neurons in the field show signs of shrinkage and apoptotic profiles.
Supplementary Figure 1.
We show also for comparison (below) the images of Hoechst staining presented in Figure S1A, but taken with shorter exposure time; in these conditions MN nuclei are barely visible, but morphology of nuclei of other cells can be clearly rated.
-I would recommend reorganizing some figures. The authors compare GL and TL and present two separate panels. It would be better combining these panels. Please, also consider the combining of panels in Fig.3 to single panel (similar for Figures 4 and 5).
RESPONSE 4.4. According to your suggestion, we combined the two panels of former Figure 2 into one panel in the new Figure 2. Regarding Figures 3 to 6, in a process of preparation of the manuscript we considered to unify the scale for all the genes in one panel. However, because some genes have much higher expression level than other genes, by using one scale in one panel we would make the genes with lower expression invisible in the graph.
Please, add the scale bar in Figure 1A and clearly indicate what green and red color mean. Why does the background in green channel (bottom line, left image) is red?
RESPONSE 4.5. In the current version of the manuscript we combined Figure 1 and Figure 10 preparing a new Figure 1, according to a suggestion of the first reviewer. We added the scale bar and included the color-coded abbreviations of TA and GL MNs to make self-explanatory the meaning of green and red color in new Figure 1C.
Regarding the reddish background in green channel we traced the parameters and conditions of image collecting; it was identified to be due to an ambient light which could not be eliminated while I was collecting the samples from a set of sections (Benjun Ji). Below (left) we present the original image taken from the LMD microscope in those circumstances; it is also a little reddish, and the background on the image you mentioned is more red owing to the increased brightness of the image used to make neuron visibility better. To avoid that issue, we replaced the image with a better image taken from the rat spinal cord from the same group, but collected in the dark (the modified Figure 1 (see a frame on the right).
-The authors indicate that AMPAR Ca2+-permeability is determined by the subunit composition and describe the role of Q/R editing in AMPAR-mediated Ca2+-inflow tuning. It is known that almost 100% of GluA2 pre-mRNA is edited in adult animals. However, ADARs expression can be changed at some pathologies, thus leading to alterations in the editing. Please, discuss it.
RESPONSE 4.6. Thank you for that valuable indication. We examined the transcript level of the Adarb1 gene, coding ADAR2 enzyme, which is essential enzyme for GluR2 pre-mRNA editing at Q/R site-607, which gates Ca2+ entry through AMPA receptor channels. The results showed (a graph below) that the mRNA level of Adarb1 was decreased both in GL and TA MNs after SCT, providing a hint that less GluR2 is edited and more Calcium ions may pass through unedited GluA2. We discussed that result (Figure S2) and possibility in the Discussion section.
The references quoted in the responses:
- Westenbroek, R. E.; Hoskins, L.; Catterall, W. A., Localization of Ca2+ channel subtypes on rat spinal motor neurons, interneurons, and nerve terminals. The Journal of neuroscience : the official journal of the Society for Neuroscience 1998, 18, (16), 6319-30.
- Jiang, M. C.; Birch, D. V.; Heckman, C. J.; Tysseling, V. M., The Involvement of CaV1.3 Channels in Prolonged Root Reflexes and Its Potential as a Therapeutic Target in Spinal Cord Injury. Front Neural Circuits 2021, 15, 642111.
- Marcantoni, M.; Fuchs, A.; Löw, P.; Bartsch, D.; Kiehn, O.; Bellardita, C., Early delivery and prolonged treatment with nimodipine prevents the development of spasticity after spinal cord injury in mice. Science translational medicine 2020, 12, (539).
- Perrier, J. F.; Hounsgaard, J., 5-HT2 receptors promote plateau potentials in turtle spinal motoneurons by facilitating an L-type calcium current. J Neurophysiol 2003, 89, (2), 954-9.
- Sukiasyan, N.; Hultborn, H.; Zhang, M., Distribution of calcium channel Ca(V)1.3 immunoreactivity in the rat spinal cord and brain stem. Neuroscience 2009, 159, (1), 217-35.
- Berthele, A.; Boxall, S. J.; Urban, A.; Anneser, J. M.; Zieglgänsberger, W.; Urban, L.; Tölle, T. R., Distribution and developmental changes in metabotropic glutamate receptor messenger RNA expression in the rat lumbar spinal cord. Brain Res Dev Brain Res 1999, 112, (1), 39-53.
- Scheefhals, N.; MacGillavry, H. D., Functional organization of postsynaptic glutamate receptors. Mol Cell Neurosci 2018, 91, 82-94.
- Alvarez, F. J.; Villalba, R. M.; Carr, P. A.; Grandes, P.; Somohano, P. M., Differential distribution of metabotropic glutamate receptors 1a, 1b, and 5 in the rat spinal cord. The Journal of comparative neurology 2000, 422, (3), 464-87.
- Hovelsø, N.; Sotty, F.; Montezinho, L. P.; Pinheiro, P. S.; Herrik, K. F.; Mørk, A., Therapeutic potential of metabotropic glutamate receptor modulators. Current neuropharmacology 2012, 10, (1), 12-48.
- Zhang, M., Normal Distribution and Plasticity of Serotonin Receptors after Spinal Cord Injury and Their Impacts on Motor Outputs. In Recovery of Motor Function Following Spinal Cord Injury, 2016.
- Perrier, J. F.; Rasmussen, H. B.; Christensen, R. K.; Petersen, A. V., Modulation of the intrinsic properties of motoneurons by serotonin. Current pharmaceutical design 2013, 19, (24), 4371-84.
- Heckman, C. J.; Lee, R. H.; Brownstone, R. M., Hyperexcitable dendrites in motoneurons and their neuromodulatory control during motor behavior. Trends in Neurosciences 2003, 26, (12), 688-695.
- Heckman, C. J.; Mottram, C.; Quinlan, K.; Theiss, R.; Schuster, J., Motoneuron excitability: the importance of neuromodulatory inputs. Clin Neurophysiol 2009, 120, (12), 2040-2054.
- Harvey, P. J.; Li, X.; Li, Y.; Bennett, D. J., 5-HT2Receptor Activation Facilitates a Persistent Sodium Current and Repetitive Firing in Spinal Motoneurons of Rats With and Without Chronic Spinal Cord Injury. Journal of Neurophysiology 2006, 96, (3), 1158-1170.
- Li, X.; Murray, K.; Harvey, P. J.; Ballou, E. W.; Bennett, D. J., Serotonin facilitates a persistent calcium current in motoneurons of rats with and without chronic spinal cord injury. J Neurophysiol 2007, 97, (2), 1236-46.
- Perrier, J. F.; Delgado-Lezama, R., Synaptic release of serotonin induced by stimulation of the raphe nucleus promotes plateau potentials in spinal motoneurons of the adult turtle. The Journal of neuroscience : the official journal of the Society for Neuroscience 2005, 25, (35), 7993-9.
- D'Amico, J. M.; Murray, K. C.; Li, Y.; Chan, K. M.; Finlay, M. G.; Bennett, D. J.; Gorassini, M. A., Constitutively active 5-HT2/alpha1 receptors facilitate muscle spasms after human spinal cord injury. J Neurophysiol 2013, 109, (6), 1473-84.
- Murray, K. C.; Stephens, M. J.; Ballou, E. W.; Heckman, C. J.; Bennett, D. J., Motoneuron excitability and muscle spasms are regulated by 5-HT2B and 5-HT2C receptor activity. J Neurophysiol 2011, 105, (2), 731-48.
- Murray, K. C.; Nakae, A.; Stephens, M. J.; Rank, M.; D'Amico, J.; Harvey, P. J.; Li, X.; Harris, R. L.; Ballou, E. W.; Anelli, R.; Heckman, C. J.; Mashimo, T.; Vavrek, R.; Sanelli, L.; Gorassini, M. A.; Bennett, D. J.; Fouad, K., Recovery of motoneuron and locomotor function after spinal cord injury depends on constitutive activity in 5-HT2C receptors. Nature medicine 2010, 16, (6), 694-700.

Round 2
Reviewer 1 Report
1) Fig.9 is mentioned twice and the text doesn't reflect anything about keeping it twice in the manuscript. If there are any specific reasons then please include them to the manuscript or keep only one of them. There is also some unedited text on Page 18/19.
2) "RESPONSE 1.3. Our previous studies showed that SCT leads to a long term decrease (5-6 weeks) in markers of synaptic nerve endings abutting motoneurons ([1] (the degree of loss – 18% of synaptophysin labeling surrounding large neurons of the ventral horn in lumbar L3/4 segments; -26% in synaptic zinc staining), a population of cholinergic [2] and excitatory (Glutamatergic) as well as inhibitory (GABAergic) terminals [3, 4]. These responses of inputs to the injury prompted us to ask whether postsynaptic receptors adapt to developing changes. In the submitted manuscript, our results showed decreased transcript level of inhibitory GABAergic and glycinergic receptors which is in line with reduced inhibitory (GABAergic) inputs and suggests that the compensatory mechanism of up-regulation is not activated. The changes in the transcript levels of glutamatergic receptors were diverse, with increased AMPAR GluA1 and mGluR1, and decreased NMDAR."
Please include in the discussion.
Author Response
Answers (in Red) to the reviewer reports:
Open review 1 (Round 2):
Comments and Suggestions for Authors:
1) Fig.9 is mentioned twice and the text doesn't reflect anything about keeping it twice in the manuscript. If there are any specific reasons then please include them to the manuscript or keep only one of them. There is also some unedited text on Page 18/19.
Responses: Thank you for these important notes. In accordance to them, we kept only one mention of Figure 9 in the Discussion section. We corrected the unedited text on Page 18/19.
2) "RESPONSE 1.3. Our previous studies showed that SCT leads to a long term decrease (5-6 weeks) in markers of synaptic nerve endings abutting motoneurons ([1] (the degree of loss – 18% of synaptophysin labeling surrounding large neurons of the ventral horn in lumbar L3/4 segments; -26% in synaptic zinc staining), a population of cholinergic [2] and excitatory (Glutamatergic) as well as inhibitory (GABAergic) terminals [3, 4]. These responses of inputs to the injury prompted us to ask whether postsynaptic receptors adapt to developing changes. In the submitted manuscript, our results showed decreased transcript level of inhibitory GABAergic and glycinergic receptors which is in line with reduced inhibitory (GABAergic) inputs and suggests that the compensatory mechanism of up-regulation is not activated. The changes in the transcript levels of glutamatergic receptors were diverse, with increased AMPAR GluA1 and mGluR1, and decreased NMDAR."
Please include in the discussion.
Responses: Thank you for the suggestion. We included this information in the Discussion section.

Reviewer 3 Report
Accept in present form
Author Response
Answers (in Red) to the reviewer reports:
Open review 3 (Round 2):
Comments and Suggestions for Authors:
Accept in present form
Responses: Thank you very much for your positive evaluation of the work.

Reviewer 4 Report
The authors have substantially revised the manuscript and added new data in accordance to the comments. Most of my comments have been addressed, so the manuscript can be recommended for the publication. I only recommend adding the information from the response to my first comment (response 4.1) into the text.
Author Response
Answers (in Red) to the reviewer reports:
Open review 4 (Round 2):
Comments and Suggestions for Authors:
The authors have substantially revised the manuscript and added new data in accordance to the comments. Most of my comments have been addressed, so the manuscript can be recommended for the publication. I only recommend adding the information from the response to my first comment (response 4.1) into the text.
Responses: Thank you very much for suggestion. We organized this information and added respective parts to the Result and Discussion sections.
